# Machine learning application for predicting smoking cessation among US adults: An analysis of waves 1-3 of the PATH study

Mona Issabakhsh[1]*, Luz Maria Sánchez-Romero[1], Thuy T. T. Le[2], Alex C. Liber[1], Jiale Tan[3], Yameng Li[1], Rafael Meza[4], David Mendez[2], David T. Levy[1]

**1** Department of Oncology, Lombardi Comprehensive Cancer Center, Georgetown University, Washington DC, United States of America, **2** Department of Health Management and Policy, University of Michigan School of Public Health, Ann Arbor, MI, United States of America, **3** Department of Epidemiology, University of Michigan School of Public Health, Ann Arbor, MI, United States of America, **4** Integrative Oncology, BC Cancer Research Institute, Vancouver, BC, Canada

* mi416@georgetown.edu

## Abstract

Identifying determinants of smoking cessation is critical for developing optimal cessation treatments and interventions. Machine learning (ML) is becoming more prevalent for smoking cessation success prediction in treatment programs. However, only individuals with an intention to quit smoking cigarettes participate in such programs, which limits the generalizability of the results. This study applies data from the Population Assessment of Tobacco and Health (PATH), a United States longitudinal nationally representative survey, to select primary determinants of smoking cessation and to train ML classification models for predicting smoking cessation among the general population. An analytical sample of 9,281 adult current established smokers from the PATH survey wave 1 was used to develop classification models to predict smoking cessation by wave 2. Random forest and gradient boosting machines were applied for variable selection, and the SHapley Additive explanation method was used to show the effect direction of the top-ranked variables. The final model predicted wave 2 smoking cessation for current established smokers in wave 1 with an accuracy of 72% in the test dataset. The validation results showed that a similar model could predict wave 3 smoking cessation of wave 2 smokers with an accuracy of 70%. Our analysis indicated that more past 30 days e-cigarette use at the time of quitting, fewer past 30 days cigarette use before quitting, ages older than 18 at smoking initiation, fewer years of smoking, poly tobacco past 30-days use before quitting, and higher BMI resulted in higher chances of cigarette cessation for adult smokers in the US.

## Introduction

Smoking prevalence in the United States (US) has significantly decreased over time (from 23.3% in 2000 [1] to 13.7% in 2018) [2]. Still, cigarette smoking continues to be one of the most significant public health issues, responsible for about 480,000 deaths annually in the US [3]. Smoking cessation is the most cost-effective means of tobacco-related disease prevention

**Data Availability Statement:** Data are from the Public Use Files for the Population Assessment of

Tobacco and Health, Waves 1-3 at https://www.icpsr.umich.edu/icpsrweb/NAHDAP/studies/36498.

**Funding:** YES, Research reported in this publication was supported by the National Cancer Institute of the National Institutes of Health (NIH) and FDA Center for Tobacco Products (CTP) under Award Number U54CA229974. The content is solely the responsibility of the authors and does not necessarily represent the official views of the NIH or the Food and Drug Administration. MI and TTTL were funded. The funders had no role in study design, data collection and analysis, decision to publish, or preparation of the manuscript.

**Competing interests:** The authors have declared that no competing interests exist.

[4]. The global and national importance of smoking cessation has been discussed widely in the literature [5–7]. To promote smoking cessation, the World Health Organization (WHO) has also emphasized strengthening its Framework Convention on Tobacco Control implementation in all countries [8]. Although most cigarette users want to quit smoking, and more than half of current smokers make a quit attempt every year, less than 10% remain abstinent for at least six months [9]. Identifying the factors driving and sustaining smoking cessation is thus a critical need to address this epidemic effectively.

Current literature on smoking cessation is dominated by population-based studies that account for a small number of predictors. Those studies often require estimating state transition rates for cessation, which makes model application and prediction more challenging [10, 11]. On the other hand, machine learning (ML) algorithms use a flexible model structure that can include multiple variables without needing state transition rates. ML has been used successfully in tobacco research for high-quality estimations and predictions and considering multiple predictors [12]. Researchers have applied ML algorithms to assist with patient smoking status classification from electronic medical records data [13]. ML models have also been used for smoking cessation prediction based on clinical data [14]. Lai et al. [15] developed ML algorithms for smoking cessation outcome prediction among current smokers enrolled in a cessation program at a medical center in Northern Taiwan. Coughlin et al. [9] used ML to predict abstinence from smoking using cognitive behavioral therapy data for tobacco dependence in the US. Medina and Mohaghegh [16] developed ML models to predict smoking cessation outcome among Quitline service users in New Zealand. These studies have mainly focused on cessation program data. Only individuals with an intention to quit participate in such programs, which limits the generalizability of the results. Applying ML to a nationally representative survey would provide, in contrast, a broad perspective on the factors that can lead to smoking cessation among the general population.

This study uses the US nationally representative longitudinal data from the Population Assessment of Tobacco and Health (PATH) [17] survey to develop ML predictive models (i.e., binary classifiers). Our objective is to analyze the smoking cessation process by distinguishing its important determinants and predicting smoking cessation after one data wave (roughly one year) for survey participants. This study should help us better understand how the transition between current and former cigarette use (a.k.a. smoking cessation) happens over time, and thereby better characterize factors that support individuals in their smoking cessation journey, both to confirm factors that have been established in the current literature, and to discover novel factors missed in previous studies. To the best of our knowledge, the PATH study has not previously been used to predict smoking cessation using ML algorithms. The outcome of this research is a set of significant determinants of smoking cessation, and accurate ML predictive models for smoking cessation, considering the nationally representative longitudinal data from the PATH survey.

## Materials and methods

### Data

We used data from the PATH survey, a nationally representative US longitudinal cohort study of tobacco use and its effects on population health [18]. We conducted a longitudinal analysis of PATH data in our study with one-time measurements. The same assumptions (as described below) are considered throughout the data cleaning and model development steps. We used the open-access PATH dataset (not the restricted version) [17], in which all data were fully deidentified. Therefore, Georgetown University and the University of Michigan Institutional Review Boards exempted our analysis from review. To develop the predictive models, we used

unweighted PATH adult survey data (ages 18 and above) from wave 1 (September 2013 to December 2014) and wave 2 (October 2014 to October 2015) [17]. We considered current cigarette smokers at wave 1 and checked whether they quit by wave 2. We only focused on PATH waves 1 and 2 to elicit variables and attributes involved in smoking cessation before the use of e-cigarettes (specifically JUUL) became more widespread, to limit the effect of changes in cigarette smoking due to e-cigarette use (and its unstable patterns in the initial years) on our results [19].

Our baseline sample included current smokers in wave 1, who smoked 100 cigarettes or more during their lifetime and reported smoking every day or some days at the time of the survey. In other words, we considered current established smokers in wave 1. We tracked the participants' smoking cessation (defined below) in wave 2. In total, 32,320 adult respondents were surveyed in PATH wave 1, among which 26,447 individuals also participated in wave 2. Out of all adults who participated in both waves, 9,281 were current established smokers in wave 1.

In this study, we applied binary classifiers to predict cigarette smoking cessation (quit or not quit). Cigarette smoking cessation is considered for those current established cigarette smokers in wave 1 who did not report smoking a cigarette in the past 30 days and self-reported quitting smoking cigarettes in wave 2 [20]. More specifically, individuals who answered "YES" to the question "Have you completely quit smoking cigarettes?" AND "NO" to the question "In the past 30 days, did you smoke a cigarette, even one or two puffs?" are considered to quit smoking cigarettes in wave 2. We excluded participants who did not answer either of these questions. Of the 9,281 current established smokers in wave 1 who participated in wave 2, 710 successfully quitted by wave 2. (Fig 1).

$$Cessation\ outcome\ (binary) : \begin{cases} Quit\ (1) \\ Not\ quit\ (0) \end{cases}$$

## Data cleaning

Data cleaning is the process of removing (or fixing) incorrect, irrelevant, corrupted, incorrectly formatted, incomplete, or duplicate data within a dataset to ensure data quality. Data cleaning is essential in model development since it speeds up the model training process, improves model accuracy and interpretability, and reduces model complexity and overfitting [21]. In our analysis, we initially considered all PATH wave 1 adult survey variables (1,742 in total), mainly related to participants' characteristics and tobacco product use habits. To this set, we

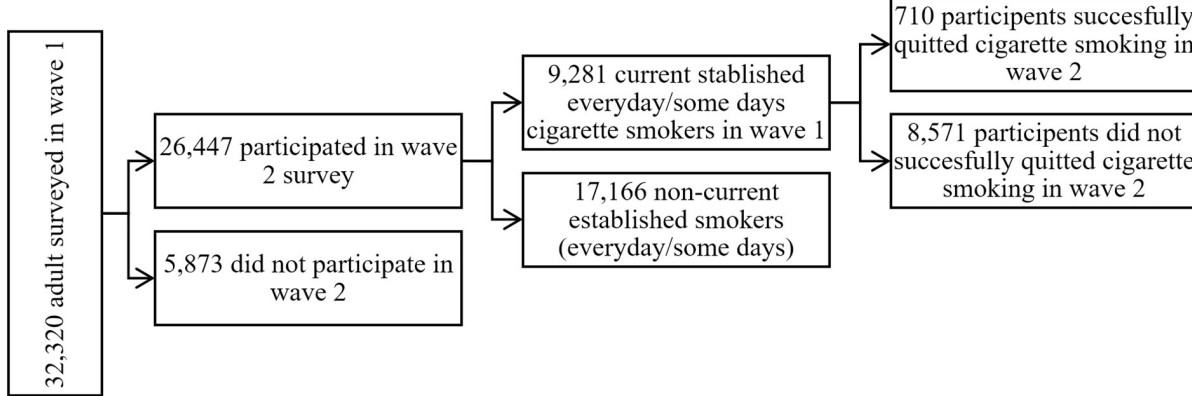

**Fig 1. Analytical sample selection flowchart, PATH adult survey waves 1 and 2.**

added 24 PATH-drived variables related to participants' past 30 days use (for different tobacco products in wave 1 and tobacco products other than cigarettes in wave 2) and their nicotine dependence, provided by The Center for the Assessment of Tobacco Regulations (CAsToR) Data Analysis and Dissemination Core [22]. We considered "tobacco products use in the past 30 days" in wave 1 to assess the relationship between participants' tobacco product use in wave 1 and cigarette quit over time in wave 2. Past 30 days use of tobacco products other than cigarettes is also considered for wave 2 to assess if the use of other tobacco products correlates with participants' cigarette quit at the time of quitting.

We started the analysis with 1,766 variables in total. In the data cleaning process, we removed variables that exclusively targeted never and former smokers, survey design variables, and categorical variables with a single level (no variation). Variables were also merged as needed. For instance, in the wave 1 PATH survey, two separate variables show the amount usually paid for a pack of cigarettes in dollars and cents. The summation of these two variables shows the total amount paid for a pack of cigarettes. In this case, to calculate a single "cigarette price" variable, we multiplied the "dollar" variable by 100 and added it to the "cents" variable.

After the initial data cleaning, we divided the remaining 1,096 variables into categories based on their similarities. From each category, we selected those variables that were most relevant to current smokers and smoking cessation, with the least number of missing samples. We then developed a correlation matrix and removed correlated variables to avoid collinearity [23, 24]. The number of "quit" instances in the dataset was limited (710 among 9,281). Therefore, as the last step of data preparation, we filled in the missing data. In other words, we added "missing" as a factor level and labeled NA samples as "missing" for categorical variables. For numeric variables, we filled in the NAs with the average value of the variable. The final set included 181 variables that were considered for the model fitting. Fig 2 shows a summary of the data cleaning process, and the details of the data cleaning are also provided in the S1 Appendix.

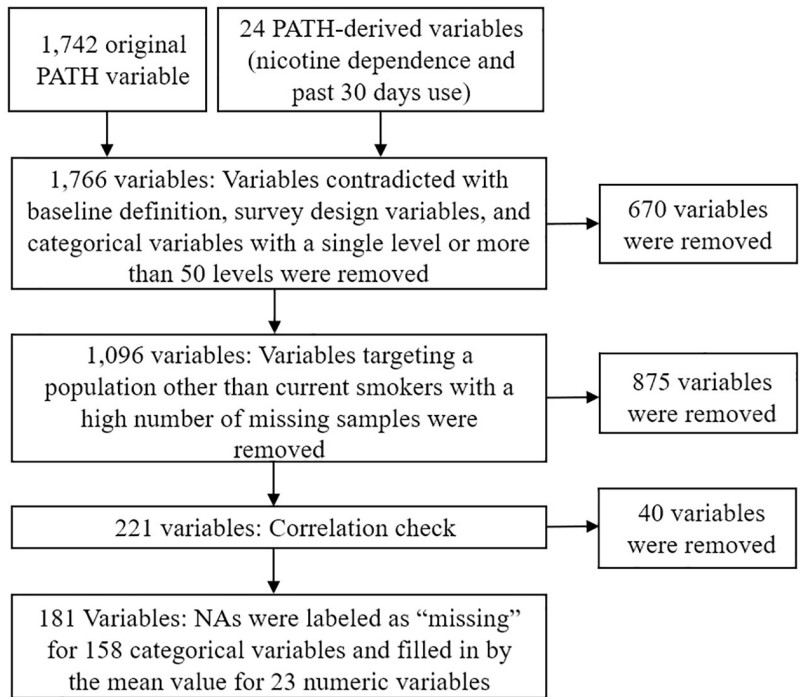

**Fig 2. The data cleaning process.**

## Class imbalance

Among all smokers in PATH wave 1, only 7% quit smoking after one wave. This issue is called class imbalance in the ML literature [25–27], and is a challenge in deploying ML algorithms for smoking cessation prediction using both clinical [28] and survey data, due to the inherent low rate of quits. Class imbalance may result in models that overweight specificity (detecting "not quits") over sensitivity (detecting "quits"), given that most individuals do not have the desired outcome (here, smoking cessation).

Advanced ML sampling techniques, including random over and under sampling and ensemble-based methods, have been developed to deal with class imbalance [28]. Random over-sampling and under-sampling methods are mainly applied to decrease the skewed class distribution effect on the performance of the classifiers [29]. In random over-sampling, samples are replicated in the minority class in the training set (i.e., quit samples are replicated). In contrast, in random under-sampling, samples are discarded from the majority class in the training set (i.e., not quit samples are discarded) to balance the class distribution [28]. In ensemble-based methods (e.g., bagging), the total data is divided into $n$ parts, and $n$ different models are trained. Each model uses all the samples of the minority class and $1/n$ of the majority class [30]. Thereby, all samples of the majority class are used for training (no information loss), while samples of the minority class are used efficiently. Because of the skewed class distribution in our data, class imbalance was a major issue. We applied random sampling and ensemble-based techniques for feature selection and predictive model training to overcome this problem.

## Variable importance and direction effect

We used ML model fitting algorithms to find the most relevant and significant variables in predicting smoking cessation among those 181 variables selected in the data cleaning step (Fig 2). To obtain significant variables arranged in the order of importance, we applied two tree-based algorithms: Random Forest (RF) [31] and Gradient Boosting Machine (GBM) [32]. Tree-based algorithms are known for handling class imbalance efficiently [28]. GBM and RF are both tree-based algorithms but differ in how the trees are built and how the results are combined. GBM builds one tree at a time; therefore, each new tree helps correct errors made by previously trained trees. RF, on the other side, trains each tree independently using a random sample of the data. RF and GBM provide variable importance, representing the average increase in prediction error or decrease in prediction accuracy when the variable is removed from the model [33]. Because of the differences between the structure of GBM and RF, the variable importance provided by these two algorithms is not completely identical. Therefore, we combined the top variables selected by both algorithms to increase the probability of identifying relevant variables of smoking cessation.

To determine the direction of the relationship between each variable and cessation, we used the SHapley Additive exPlanation (SHAP) [34] technique. SHAP is a conditional game theory method that explains instances' predictions by computing each variable's contribution (and the effect direction of its contribution) to the prediction. We use TreeSHAP [35], a variant of SHAP for tree-based ML algorithms, to determine the effect direction of each variable on cigarette cessation. The TreeSHAP is a typical method for variable interpretation, specifically in the public health domain. It has been used in tobacco research [16], cancer prevention and control research [36], and other applications [37]. The TreeSHAP analysis is used to explain the prediction of the machine learning models independently by each variable included in the model.

## Machine learning predictive models

After acquiring the analytical sample, variables were selected to be included in the classification model. The sample was then divided into a training (70%) and a testing (30%) dataset

[38]. The training dataset was used to "train" the classifiers to predict the smoking cessation for each individual. The testing dataset (which shared the same distribution and properties as the training dataset) was not used in the training process. After training, the testing dataset was used to evaluate the ability of each classifier to predict smoking cessation.

We developed classification models with Generalized Linear Regression (GLM) (an extension of the linear regression for binary classification) [39], RF [31], GBM [32], and extreme gradient boosting (XGBoost) [40] algorithms. We assessed four sampling strategies (no sampling, under-sampling, over-sampling, and bagging) in the classifiers' training to overcome the class imbalance issue and to evaluate the effect of the sampling strategy on the performance of the classifiers in the testing set. The training process of all four algorithms was performed in the R statistical software version 4.0.2, using the training set data (70% of the total analytical samples) [41]. The hyperparameters of algorithms were adjusted by experimenting to achieve the best performance for the testing set. By limiting the complexity of each model, the likelihood of overfitting was reduced.

The classifiers were evaluated based on the ability to predict cases correctly identified as "quit", compared with cases incorrectly identified as "'quit", and cases correctly identified as "not quit", compared with cases incorrectly identified as "not quit" [42]. The performance of the trained classifiers was compared based on classification accuracy (the ability to make correct predictions), sensitivity (the ability to predict "quit" cases correctly), specificity (the ability to predict "not quit" cases correctly), and the area under the receiver operating characteristic curve (AUC-ROC), (the ability to make correct predictions) in the testing set [42].

## Results

### Variable selection outcome

Out of 181, GBM selected 67, and RF selected 178 significant variables. The difference between the number of variables selected by GBM and RF is due to different algorithm structures (as discussed in the previous section). GBM places significant weight on the top 12 variables and 1% weight on the rest, while RF distributes the importance weight between multiple variables. Table 1 shows the variables in order of importance determined by GBM and RF. As shown in Table 1, 80% of the top 15 variables (72% of the top 25) selected by RF and GBM are the same, showing robustness of results across the two variable selection algorithms. The importance order, however, is different between GBM and RF.

Based on the GBM results, the most significant variable in smoking cessation prediction in wave 2 is the age at smoking initiation, followed by e-cigarette use in the past 30 days in wave 2 and poly tobacco product use in wave 1. Next was the number of years having smoked cigarettes, followed by the participant's body mass index (BMI). In general, out of 67 variables detected significant by GBM, 11 variables (16%) correspond to individual's cigarette smoking history and cigarette use habits in wave 1, 11 (16%) are variables related to tobacco products use (other than cigarettes and e-cigarettes) in waves 1 and 2, 10 (15%) are physical and mental health and quality of life variables in wave 1, 9 (13%) are demographics and socio-economic status (SES) variables in wave 1, 7 (10%) are variables related to family and friends tobacco products use habits and rules about tobacco products use at home in wave 1, 5 (8%) are variables related to knowledge and beliefs about smoking and tobacco products harmfulness in wave 1, 4 (6%) are variables related to the exposure to tobacco ads and promotions in wave 1, 3 (5%) are alcohol and other substances use variables in wave 1, 3 (5%) are variables related to noticing tobacco products health warnings in wave 1, 2 (3%) are e-cigarette use variables in waves 1 and 2, and 2 (3%) are variables related to social media exposure in wave 1.

**Table 1. Comparison of the top 25 variables selected by GBM and RF.**

| No. | Variable | Category | GBM order | RF order |
|---|---|---|---|---|
| 1 | Age range when first started smoking cigarettes every day | Cigarette smoking habits | 1 | 3 |
| 2 | In the past 30 days, the number of days used e-cigarettes | Wave 2 past 30-days use | 2 | 7 |
| 3 | Adult poly tobacco product user (used at least 10 days in the past 30 days) | Wave 1 past 30-days use | 3 | 6 |
| 4 | How long smoked cigarettes fairly regularly | Cigarette smoking habit | 4 | 1 |
| 5 | Body mass index (BMI) | Demographics | 5 | 2 |
| 6 | In the past 30 days, the number of days smoked cigarettes | Wave 1 past 30-days use | 6 | 4 |
| 7 | Number of minutes from waking to smoking the first cigarette | Cigarette smoking habit | 7 | 5 |
| 8 | In the past 30 days, the number of days smoked cigarillos | Wave 2 past 30-days use | 8 | 58 |
| 9 | How would you describe your overall opinion of tobacco | Beliefs | 9 | 15 |
| 10 | The extent to which health warnings on cigarette packs make you more likely to quit/stay quit from smoking | Health warnings | 10 | 13 |
| 11 | In the past 30 days, the number of days smoked filtered cigars | Wave 2 past 30-days use | 11 | 113 |
| 12 | Age range when interviewed | Demographics | 12 | 8 |
| 13 | How often have you seen a list of the chemicals contained in tobacco products in the past 12 months | Health warnings | 13 | 14 |
| 14 | General perception: Harmfulness of cigarettes to health | Beliefs | 14 | 22 |
| 15 | How often do you use the internet | Social media | 15 | 11 |
| 16 | Number of hours in the past 7 days that you were in close contact with others when they were smoking | Family/friends smoking habits | 16 | 9 |
| 17 | In the past 30 days, the average number of cigarettes smoked per day on days smoked | Wave 1 past 30-days use | 17 | 42 |
| 18 | Statement that best describes rules about using non-combustible tobacco products inside your home | Smoking rules at home | 18 | 45 |
| 19 | Self-perception of quality of life | Physical/mental health and quality of life | 19 | 23 |
| 20 | Current employment status | SES | 20 | 43 |
| 21 | Currently covered by health insurance or health coverage plan | SES | 21 | 25 |
| 22 | Statement that best describes rules about smoking a combustible tobacco product inside your home | Smoking rules at home | 22 | 41 |
| 23 | Last time that you used alcohol or other drugs weekly or more often | Alcohol/other substances use | 23 | 21 |
| 24 | Opinion on using tobacco among people who are important to you | Family/friends smoking habits | 24 | 16 |
| 25 | Used a coupon when buying cigarettes in the past 30 days | Cigarette smoking habits | 25 | 36 |
| 26 | The amount usually paid for a pack of cigarettes | Cigarette smoking habits | 26 | 10 |
| 27 | How often have you thought about the chemicals contained in tobacco in the past 12 months | Health warnings | 35 | 18 |
| 28 | Highest grade or level of school completed | SES | 38 | 12 |
| 29 | Level of satisfaction with social activities and relationships | Physical/mental health and quality of life | 39 | 24 |
| 30 | How often you have been bothered by emotional problems such as feeling anxious, depressed, or irritable in the past 7 days | Physical/mental health and quality of life | 41 | 20 |
| 31 | How often have you noticed things that promote tobacco in the past 6 months? | Ads/promotions | 49 | 19 |
| 32 | Self-perception of mental health | Physical/mental health and quality of life | 61 | 17 |

The top five variables selected by RF are years of smoking cigarettes, BMI, age at smoking initiation, past 30 days poly tobacco product use in wave 1, and minutes from waking up to smoke the first cigarette in wave 1. In general, out of 178 variables detected significant by RF, 64 (35%) correspond to tobacco products use (other than cigarettes and e-cigarettes) in waves 1 and 2, 35 (20%) are physical and mental health and quality of life variables in wave 1, 16 (9%) are variables regarding knowledge and beliefs about smoking and tobacco products harmfulness in wave 1, 14 (8%) are demographic and SES variables in wave 1, 14 (8%) are related to individual's cigarette smoking history and cigarette use habits in wave 1, 8 (4%) are variables

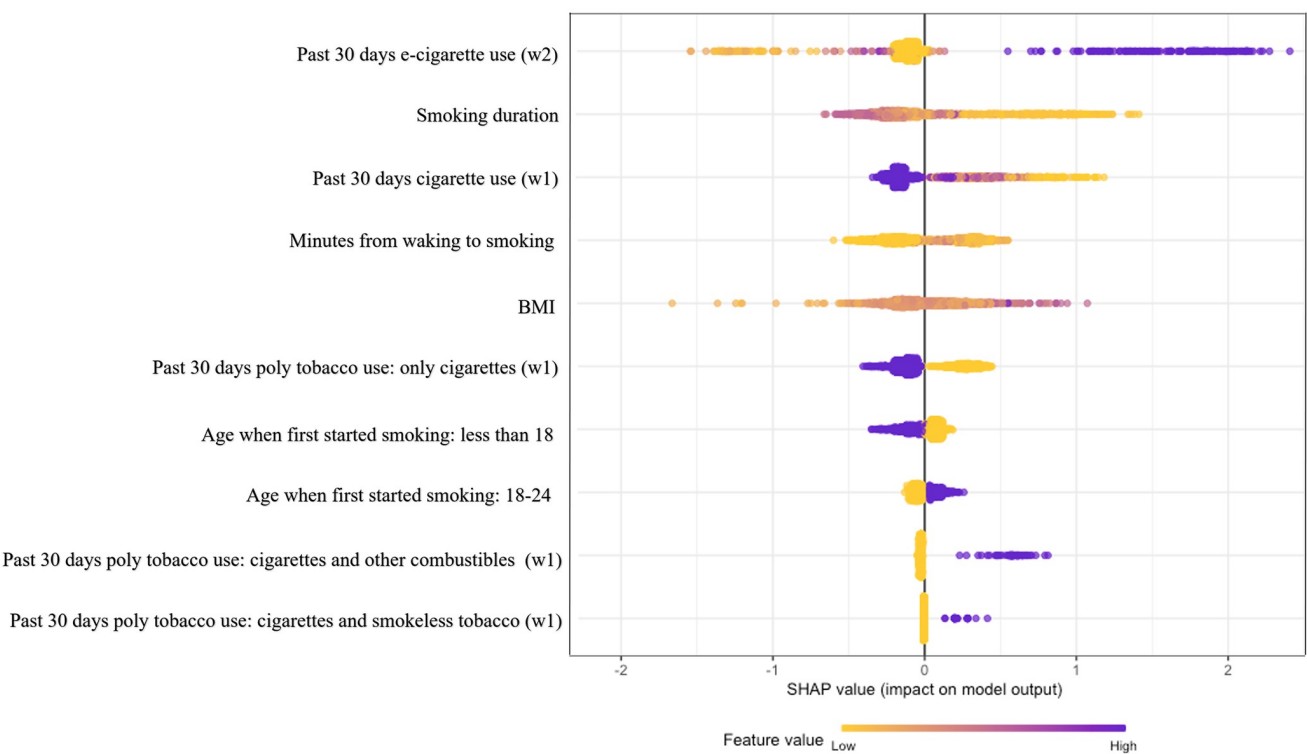

**Fig 3. The TreeSHAP summary plot for the combination of the top five variables selected by RF and GBM.**

related to alcohol and other substances use in wave 1, 7 (4%) are e-cigarette use variables in waves 1 and 2, 7 (4%) are variables related to family and friends tobacco products use habits and rules about tobacco products use at home in wave 1, 5 (3%) are variables regarding exposure to tobacco ads and promotions in wave 1, 3 (2%) are social media exposure variables, 3 (2%) are variables related to noticing tobacco products health warnings in wave 1, and 2 (1%) are variables about previous quit attempts.

We selected the combination of the top 25 variables detected significant by both GBM and RF algorithms shown in Table 1 to develop our final model to predict smoking cessation for participants in wave 2. The final dataset included 32 variables.

Fig 3 shows the TreeSHAP summary plot for the combination of the top five variables selected by RF and GBM. Each point on the summary plot is a SHAP value for a variable and a survey participant, equivalent to the variable's marginal contribution to the cessation prediction. In the summary plot in Fig 3, the y-axis shows variable names, while the x-axis shows the SHAP values. In the plot, colors show the original value of each variable. For categorical variables, two colors (yellow and dark purple), are shown to compare two levels of the variable. In contrast, for continuous variables, a spectrum of colors (from yellow to dark purple) is shown to demonstrate different values of the variable. SHAP values less than zero show "low predicted cessation," while values greater than zero show "high predicted cessation."

Based on the summary plot, we observed that lower values of past 30 days e-cigarette use in wave 2 mainly resulted in lower predicted cessation, while higher values of past 30 days e-cigarette use in wave 2 resulted in higher predicted cessation. More years of smoking mainly resulted in lower predicted cessation, while lower smoking duration resulted in higher predicted cessation. More past 30 days cigarette use in wave 1 resulted in lower predicted cessation, while fewer past 30 days cigarette use in wave 1 resulted in higher predicted cessation. The summary plot

showed no noticeable direction effect for minutes from waking up to smoking, which is likely due to the low variability of time to the first cigarette in the morning in our dataset. In the wave 1 PATH survey, 75% of participants reported smoking the first cigarette of the day within one hour of waking up. This can also be observed in the summary plot, where most dots are yellow for the minutes from waking up to smoking, indicating low values of this variable.

It can be observed that higher values of BMI (darker dots) resulted in higher predicted cessation, while lower values resulted in lower predicted cessation. For categorical variables, one level of the variable is compared to the other levels. The summary plot suggests that only cigarette use resulted in lower predicted cessation compared to poly tobacco products use. Alternatively, simultaneous use of cigarettes and other combustibles, and cigarettes and smokeless tobacco products in wave 1 resulted in a higher predicted cessation compared to only cigarette use. A lower predicted cessation is observed for participants who started smoking when they were 18 or younger, and comparatively, a higher predicted cessation for people who started smoking at ages 18–24. It should be noted that most participants (52%) indicated 18 or younger as the age of smoking initiation, followed by 18–24 (41%) and other ages (7%). Therefore, it can be concluded that the probability of cessation is lower for participants who started smoking at ages 18 or younger compared to older ages.

## Model performance

After model training, we used the testing dataset for the performance evaluation. We trained predictive models applying RF, GBM, GLM, and XGBoost algorithms with variables shown in Table 1. We tried different sampling strategies (no sampling, under-sampling, over-sampling, and bagging) to evaluate the effect of sampling on the prediction power of the classifiers in the test set. A comparison between the sensitivities, specificities, accuracies, and AUC-ROC values of the models for the testing set is shown in Table 2. The ROCs of the models are compared in Fig 4.

Without a sampling strategy, the classifier ignores the minority class and labels almost all samples as "not quit" because of class imbalance. Consequently, the sensitivity measure (which shows the power of the classifier to predict quit cases correctly) is close to 0, and the specificity measure (which indicates the ability of the classifier to predict not quit cases correctly) is close to 1 in all cases with no sampling. Without any sampling, balanced accuracy is also as low as 0.5 (random guess) in all cases, which again indicates the poor performance of the classifiers in predicting the actual smoking cessation outcome. Even though the AUC-ROC is around 0.75 for all cases with no sampling, it does not imply good performance. The AUC-ROC metric is high in cases with no sampling only because 93% of the data points are "not quit," which is detected perfectly well by the classifiers (specificity $\sim$ 1), neglecting 7% of "quit" cases.

With random over-sampling, classifiers performed better in sensitivity and balanced accuracy compared with no sampling. A limitation of random over-sampling is that, since the minority class samples are duplicated in the train set, not much variance is added to help with the learning process. Random under-sampling helped increase the sensitivity and balanced accuracy better than random over-sampling and much better than no sampling. However, the drawback of random under-sampling is information loss, meaning that a part of the majority class data is removed to develop a balanced training set. All classifiers performed well with bagging. Bagging resulted in sensitivity, specificity, and accuracy of around 0.7 and a mean AUC-ROC of 0.75. Compared with under and over-sampling, in bagging, all samples are used efficiently for training, with no information loss.

Considering all performance metrics shown in Table 2, all four classification algorithms performed poorly without sampling with over-sampling GBM performed best, with under-

**Table 2. Evaluation results of the predictive models.**

| Sample | Model | Sensitivity | Specificity | Balanced Accuracy | ROC-AUC |
|---|---|---|---|---|---|
| No Sampling | | | | | |
| | GBM | 0.0135 | 0.9972 | 0.5054 | 0.7696 |
| | XGBoost | 0.0676 | 0.9917 | 0.5296 | 0.7574 |
| | GLM | 0.0495 | 0.9929 | 0.5212 | 0.7392 |
| | RF | 0.0045 | 0.9992 | 0.5018 | 0.7584 |
| Over Sampling | | | | | |
| | GBM | 0.6712 | 0.7732 | 0.7222 | 0.7757 |
| | XGBoost | 0.3108 | 0.9094 | 0.6101 | 0.7021 |
| | GLM | 0.6531 | 0.7165 | 0.6848 | 0.7244 |
| | RF | 0.0360 | 0.9948 | 0.5154 | 0.7614 |
| Under Sampling | | | | | |
| | GBM | 0.7162 | 0.7114 | 0.7138 | 0.7652 |
| | XGBoost | 0.7432 | 0.6937 | 0.7185 | 0.7645 |
| | GLM | 0.6667 | 0.6409 | 0.6538 | 0.6991 |
| | RF | 0.7432 | 0.6917 | 0.7175 | 0.7652 |
| Bagging | | | | | |
| | GBM | 0.6824 | 0.7445 | 0.7135 | 0.7631 |
| | XGBoost | 0.7008 | 0.7019 | 0.7014 | 0.7557 |
| | GLM | 0.6607 | 0.6637 | 0.6622 | 0.7063 |
| | RF | 0.7297 | 0.7146 | 0.7221 | 0.7637 |

sampling GBM and RF performed best, and with bagging RF performed best. Generally, GBM, XGBoost, and RF performed better than GLM in all metrics and considering all sampling techniques.

For validation, we used the best-performing model (RF with bagging) with a similar combination of variables to predict the smoking cessation of wave 2 adult current established smokers (ages 18 and above) in wave 3. The model predicted the smoking cessation outcome for individuals in wave 3 with an accuracy level of 70% in the test set. The detailed steps of the validation process are explained in the S2 Appendix.

## Discussion

To the best of our knowledge, the PATH survey data has not previously been used to predict smoking cessation using ML predictive models, and our study is the first that surveyed all variables of at least one wave of the PATH dataset for smoking cessation prediction. Despite the class imbalance issue and a highly skewed class distribution (7% quit, 93% not quit cases), we detected important determinants of smoking cessation introduced in Table 1 and developed predictive models using these variables. No existing study in the literature has considered simultaneously all the final variables listed in Table 1 to predict smoking cessation. Our best model (RF with bagging) showed good performance in predicting smoking cessation in a representative population of adult cigarette users in the US (sensitivity 73%, specificity 71%, balanced accuracy 72%, and AUC-ROC 76%).

Our analysis showed that variables such as the age at smoking initiation, years of smoking cigarettes, BMI of participants, past 30 days use of poly tobacco products in wave 1, minutes from waking up to smoking the first cigarette in wave 1, and past 30 days use of e-cigarettes in wave 2 are important determinants of smoking cessation. We applied the TreeSHAP algorithm to assess the relationship direction between those variables and smoking cessation. The

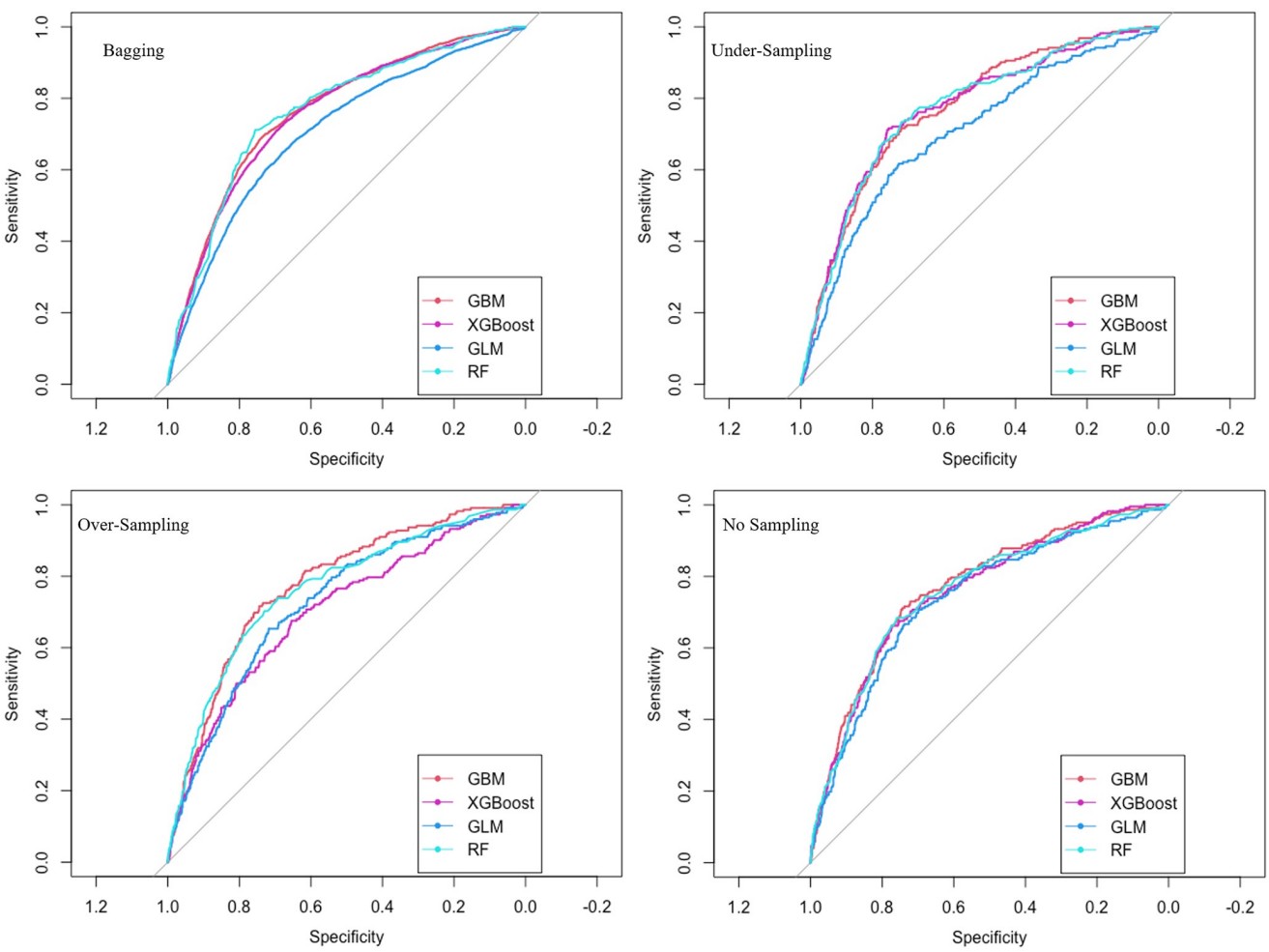

**Fig 4. ROC comparison of the predictive models.**

TreeSHAP analysis demonstrated that more e-cigarette use in the past 30 days at the time of quitting, being older than 18 at smoking initiation, smoking for fewer years, fewer cigarette use in the past 30 days before quitting, poly tobacco past 30 days use (compared with only cigarettes use) before quitting, and higher BMI mainly resulted in higher odds of cessation for adults. We use the term "mainly" since each variable affects each participant differently (dots in Fig 3). What we reported here is the effect of each variable on most participants.

Our study is consistent with previous research that has suggested an association between early smoking initiation and longer duration of smoking [15], and increased chances of nicotine dependence [15, 16] and lower chances of cessation [43, 44]. Our analysis also showed a positive association between higher BMI and cigarette quitting. This relationship could be explained by the positive correlation between higher BMI and health risks, specifically heart disease [45], which can motivate people to quit smoking cigarettes [46]. Studies have mainly discussed the effect of smoking cessation on weight gain [44]. Another ML study considered BMI as a determinant of smoking cessation but did not discuss the direction of the relationship [15]. Previous literature has reported mixed outcomes regarding the effect of e-cigarette use on smoking cessation, ranging from no effect [47] to a significant and positive effect [48, 49]. Our analysis suggested that e-cigarette use is associated with a higher chance of cessation for adult

cigarette smokers. Our study also showed that the past 30-days use of other tobacco products (both combustible and smokeless) in addition to cigarettes could increase the chance of cigarette cessation for adult smokers, compared to only cigarette use. This can be interpreted by arguing that the combination of cigarette use with other tobacco products may result in partial or complete substitution of cigarettes for adult smokers and, therefore, higher chances of cigarette quitting [50]. Our analysis did not show an obvious direction for the effect of "minutes from waking up to smoking the first cigarette" on cigarette cessation since 75% of participants in our study reported smoking the first cigarette of the day within one hour of waking up. A shorter time to the first cigarette in the morning is known in the literature as an indicator of nicotine dependence, which can reduce the chance of cessation [51]. A low degree of variability in the time between waking up and smoking the first cigarette, reported by study participants, might have affected the ability of the model to identify the direction of this relationship.

Our results point to the potential importance of non-tobacco behavioral and socioeconomic variables that are generally overlooked in previous smoking cessation studies, such as health insurance coverage [52], and mental health status [53]. Our study also revealed uncommon variables associated with smoking cessation, such as internet use and perceived quality of life. Using the internet might expose individuals to tobacco coupons and advertisements, as well as raise awareness regarding the harmfulness of tobacco through surfing websites. While many studies surveyed the effect of cigarette use on quality of life [54–56], we did not find any research exploring the effect of quality of life on smoking cessation.

Our study results show that to compare the performance of the classifiers in case of class imbalance, all metrics of sensitivity, specificity, balanced accuracy, and AUC-ROC should be considered and that only focusing on a single metric can be misleading. For instance, Fig 4 shows an almost equal performance for all classifiers; however, according to the results provided in Table 2, those classifiers with "No-sampling" performed poorly in detecting quit cases (low sensitivities). Considering a combination of all metrics in Table 2, an RF model with bagging outperformed all other classifiers. Compared with random under and over-sampled data, bagging enables using the entire train set efficiently for training the classifiers. The developed RF model with bagging predicted wave 1 (wave 2) current established smokers' cessation in wave 2 (wave 3) with a 72% (70%) accuracy level in the testing set.

Our model performed better than or equal to similar studies in the literature in predicting smoking cessation. The artificial neural network algorithm developed by Lai et al. [15] predicted smoking cessation with an average accuracy of 64%. Coughlin et al. [9] applied a classification and regression tree algorithm which predicted the smoking cessation outcome with an accuracy level of 64%. Medina and Mohaghegh [16] developed a CatBoost algorithm capable of predicting smoking cessation at 70% accuracy. However, it should be noted that because of the differences between study designs and the data used, these studies cannot be compared directly.

Our study is subject to limitations. We have mainly considered waves 1–2 (developing cohort) of the PATH survey for model development and waves 2–3 (validation cohort) for model validation. Thus, our results and conclusions may not apply to later years when other factors (such as the use of JUUL and nicotine pouches) might be more relevant. Furthermore, the validation cohort in our study had 690 fewer (7.4% lower) participants than the developing cohort. This reduction is partially because some current smokers in Wave 1 had quit by Wave 2. Additionally, based on the PATH survey user guide [17], some participants surveyed in PATH wave 1 were permanently or temporarily ineligible to participate in the follow-up waves (for instance, because they were deceased or moved out of the US). A number of eligible participants did not agree to participate in follow-up waves, and some did not respond to follow-up surveys. Even with the observed loss to follow-up between waves 1–2 and 2–3, we had a large

enough sample (w1–2: 9,281, w2–3: 8,591) to accomplish our analysis and train, test, and validate accurate predictive models. In addition, our analysis is based on PATH data for the US, and our results may not apply to specific sub-populations within the US (e.g., racial/ethnic groups) or other countries. Further validation would be necessary to assess the performance of our model in other populations or in more recent years (e.g., due to the introduction and rapid growth of JUUL in the US). Another limitation of our analysis is the inability to detect the direction effect of "minutes from waking up to smoking the first cigarette" (an important indicator of nicotine dependence) on cigarette cessation. Most participants in our baseline sample reported smoking the first cigarette of the day within one hour of waking up, which resulted in low variability of minutes from waking up to smoking the first cigarette and possibly the inability of the model to detect its direction effect.

## Conclusions

To better characterize factors that support individuals in their smoking cessation journey, and to inform future tobacco policies, we applied ML to interpret each variable's effect on smoking cessation odds. Compared to other studies which are focused on cessation treatment program data, we used data from a US nationally representative survey to predict smoking cessation among the general population, and considered a broad range of factors that could lead to smoking cessation. This study shows the virtue of ML algorithms over other modeling strategies to find important determinants of smoking cessation and develop accurate predictive models, specifically in large datasets with vast numbers of variables.

## Supporting information

**S1 Appendix. Steps of data cleaning.**
(PDF)

**S2 Appendix. Model validation for cessation transition between waves 2-3.**
(PDF)

## Acknowledgments

We would like to thank the Center for the Assessment of Tobacco Regulations (CAsToR) Data Analysis and Dissemination Core for providing data analysis insights for this work and to members of CAsToR for providing comments on the initial draft.

## Author Contributions

**Conceptualization:** Mona Issabakhsh, Rafael Meza, David Mendez, David T. Levy.

**Formal analysis:** Mona Issabakhsh.

**Funding acquisition:** Mona Issabakhsh, Thuy T. T. Le.

**Methodology:** Mona Issabakhsh.

**Software:** Mona Issabakhsh.

**Validation:** Mona Issabakhsh.

**Visualization:** Mona Issabakhsh.

**Writing – original draft:** Mona Issabakhsh, Luz Maria Sánchez-Romero, David T. Levy.

**Writing – review & editing:** Mona Issabakhsh, Thuy T. T. Le, Alex C. Liber, Jiale Tan, Yameng Li, Rafael Meza, David Mendez, David T. Levy.

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
