## [Decision Letter · Decision Letter 0]

11 Apr 2023

PONE-D-23-00290Machine learning application for predicting smoking cessation among US adultsPLOS ONE

Dear Dr. Issabakhsh,

Thank you for submitting your manuscript to PLOS ONE. After careful consideration, we feel that it has merit but does not fully meet PLOS ONE’s publication criteria as it currently stands. Therefore, we invite you to submit a revised version of the manuscript that addresses the points raised during the review process.

We look forward to receiving your revised manuscript.

Kind regards,

Mohammad Amin Fraiwan

Academic Editor

PLOS ONE

Journal Requirements:

"NO authors have competing interests."

Reviewers' comments:

Reviewer's Responses to Questions

**Comments to the Author**

1. Is the manuscript technically sound, and do the data support the conclusions?

Reviewer #1: Yes

Reviewer #2: Yes

2. Has the statistical analysis been performed appropriately and rigorously? 

Reviewer #1: Yes

Reviewer #2: Yes

3. Have the authors made all data underlying the findings in their manuscript fully available?

Reviewer #1: Yes

Reviewer #2: Yes

4. Is the manuscript presented in an intelligible fashion and written in standard English?

Reviewer #1: Yes

Reviewer #2: Yes

5. Review Comments to the Author

Reviewer #1: The manuscript is well written and the methodology section is detailed and particularly data cleaning sub-section. The limitations of the study can be described more in detail. The potential strengths of the study stand out. The study will be interesting for wider readership and adds to existing knowledge base in tobacco research.

Reviewer #2: Review Report

Title: Machine learning application for predicting smoking cessation among US adults.

Manuscript Number:

Review Version: I

Review Comments

The objective and the results are little beat not consistent.

Establish the background more and incorporate global and national promises.

Use of exclamations in inappropriate way.

There is high drop out among the first and the second wave and between the second and the third wave? What was the reason behind and how was that treated?

The inclusion criteria are loose and is that only current or life time smoking or both?

Dis TreeSHAP validated on Artificial intelligence before its application on human being?

What are the assumptions taken in to account in general and in the final model? Is that one time or repeated time measurements? The analysis should be linear regression or longitudinal analysis? How did you ensure data quality?

How did the ethical considerations secure? Did the national or local IRB approval?

The analysis needs further explanation.

Ensure the completeness of the contents of the manuscript E.g. The title lacks time period.

Regards,

6. PLOS authors have the option to publish the peer review history of their article (what does this mean?). If published, this will include your full peer review and any attached files.

Reviewer #1: No

Reviewer #2: No

---

## [Author Response · Author response to Decision Letter 0]

24 May 2023

Responses for reviewers’ comments: PLOS ONE - Manuscript ID PONE-D-23-00290

Title: Machine learning application for predicting smoking cessation among US adults

We appreciate the opportunity to revise our manuscript. We responded to the reviewers’ comments below and tracked changes in the manuscript. Please note that the page and line numbers provided in the responses refer to the “tracked changes” version of the manuscript.

Reviewer 1

The manuscript is well written and the methodology section is detailed and particularly data cleaning sub-section. The limitations of the study can be described more in detail. The potential strengths of the study stand out. The study will be interesting for wider readership and adds to existing knowledge base in tobacco research.

Response: We thank the reviewer for their valuable comment. In response to this comment, we have described the limitations of our study in more detail and revised the last paragraph of the “Discussion” section (page 11, lines 394-417) as follows:

Our study is subject to limitations. We have mainly considered waves 1-2 (developing cohort) of the PATH survey for model development and waves 2-3 (validation cohort) for model validation. Thus, our results and conclusions may not apply to later years when other factors (such as the use of JUUL and nicotine pouches) might be more relevant. Furthermore, the validation cohort in our study had 690 fewer (7.4% lower) participants than the developing cohort. This reduction is partially because some current smokers in Wave 1 had quit by Wave 2. Additionally, based on the PATH survey user guide,[1] some participants surveyed in PATH wave 1 were permanently or temporarily ineligible to participate in the follow-up waves (for instance, because they were deceased or moved out of the US). A number of eligible participants did not agree to participate in follow-up waves, and some did not respond to follow-up surveys. Even with the observed loss to follow-up between waves 1-2 and 2-3, we had a large enough sample (w1-2: 9,281, w2-3: 8,591) to accomplish our analysis and train, test, and validate accurate predictive models. 

In addition, our analysis is based on PATH data for the US, and our results may not apply to specific sub-populations within the US (e.g., racial/ethnic groups) or other countries. Further validation would be necessary to assess the performance of our model in other populations or in more recent years (e.g., due to the introduction and rapid growth of JUUL in the US). Another limitation of our analysis is the inability to detect the direction effect of “minutes from waking up to smoking the first cigarette” (an important indicator of nicotine dependence) on cigarette cessation. Most participants in our baseline sample reported smoking the first cigarette of the day within one hour of waking up, which resulted in low variability of minutes from waking up to smoking the first cigarette and possibly the inability of the model to detect its direction effect.

Reviewer 2

Comment 1: The objective and the results are little beat not consistent.

Response: We thank the reviewer for noting this. We have revised the objective sentence in the “Introduction” section (page 2, lines 33-37) as follows to make sure of the consistency of the objective and results: 

This study uses the US nationally representative longitudinal data from the Population Assessment of Tobacco and Health (PATH) survey to develop ML predictive models (i.e., binary classifiers). Our objective is to analyze the smoking cessation process by distinguishing its important determinants and predicting smoking cessation after one data wave (roughly one year) for survey participants. 

Comment 2: Establish the background more and incorporate global and national promises.

Response: We thank the reviewer for their comment. We have extended the introduction of the paper to cite studies discussing the global and national importance of smoking cessation.[2-5] We added the following sentences to the “Introduction” section (page 2 lines 6-9):

The global and national importance of smoking cessation has been discussed widely in the literature.[3-5] To promote smoking cessation, the World Health Organization (WHO) has also emphasized strengthening its Framework Convention on Tobacco Control implementation in all countries.[1]

Comment 3: Use of exclamations in inappropriate way.

Response: We thank the reviewer for noting this problem. Probably this happened because of a “compiling” issue. We have revised the manuscript and made corrections where needed. 

Comment 4: There is high drop out among the first and the second wave and between the second and the third wave? What was the reason behind and how was that treated?

Response: We thank the reviewer for raising this issue. The validation cohort (waves 2-3) in our study had 690 fewer participants (7.4% lower) than the developing cohort (waves 1-2). This reduction is partially because some current smokers in wave 1 had quit by wave 2. In addition, the PATH survey user guide explains part of this "drop out" between the baseline wave of PATH (wave 1) and the follow-up waves.[1] Based on “Section 4. Response Rates”, on page 32 of the PATH user guide, “Some addresses sampled for the PATH Study could not be located or accessed, others were found to be ineligible (e.g., vacant lots and group quarters), and some eligible households did not complete the household screener. Further, not all sampled persons within eligible households agreed to participate in the PATH Study, and those who were recruited at Wave 1, i.e., those in the Wave 1 Cohort, may not have responded at some or all of the follow-up waves”. Additionally, as explained in Section 4 of the PATH user guide, some participants surveyed in PATH wave 1 were permanently or temporarily ineligible to participate in the follow-up waves because they were deceased or moved out of the US.[1] 

Even with the observed loss to follow-up between waves 1-2 and 2-3, we had a large enough sample (w1-2: 9,281, w2-3: 8,591) to accomplish our analysis and train, test, and validate accurate predictive models, as discussed in the “Results” section. We added the following paragraph to the last part of the “Discussion” section (page 11, lines 398-407) and addressed this issue as one limitation of our study:

Furthermore, the validation cohort in our study had 690 fewer (7.4% lower) participants than the developing cohort. This reduction is partially because some current smokers in Wave 1 had quit by Wave 2. Additionally, based on the PATH survey user guide,[1] some participants surveyed in PATH wave 1 were permanently or temporarily ineligible to participate in the follow-up waves (for instance, because they were deceased or moved out of the US). A number of eligible participants did not agree to participate in follow-up waves, and some did not respond to follow-up surveys. Even with the observed loss to follow-up between waves 1-2 and 2-3, we had a large enough sample (w1-2: 9,281, w2-3: 8,591) to accomplish our analysis and train, test, and validate accurate predictive models.

Comment 5: The inclusion criteria are loose and is that only current or life time smoking or both?

Response: We thank the reviewer for their question. As explained in the manuscript, “Our baseline sample included current established cigarette smokers in wave 1, defined as those who smoked 100 cigarettes or more in their lifetime and reported smoking every day or some days.” In other words, we have considered both criteria: current smokers who smoked 100 cigarettes or more in their lifetime. To make the definition clearer, we revised the “Data” section (page 3, lines 70-73) as follows: 

Our baseline sample included current smokers in wave 1, who smoked 100 cigarettes or more during their lifetime and reported smoking every day or some days at the time of the survey. In other words, we considered current established smokers in wave 1.

Comment 6: Is TreeSHAP validated on Artificial intelligence before its application on human being?

Response: We thank the reviewer for this question. The TreeSHAP is a typical method for variable interpretation, specifically in the public health domain. TreeSHAP has been used in tobacco research,[6] in cancer prevention and control research,[7] and in other applications [8] in a similar manner to our analysis. To explain the application of TreeSHAP more, we added the following sentence to the last paragraph of the “Variable importance and direction effect” section (page 5, lines 157-161): 

The TreeSHAP is a typical method for variable interpretation, specifically in the public health domain. It has been used in tobacco research,[6] cancer prevention and control research,[7] and other applications.[8] The TreeSHAP analysis is used to explain the prediction of the machine learning models independently by each variable included in the model. 

Comment 7: What are the assumptions taken in to account in general and in the final model? Is that one time or repeated time measurements? The analysis should be linear regression or longitudinal analysis? How did you ensure data quality?

Response: We thank the reviewer for their questions. We have explained the assumptions of our model extensively in the Material and methods section and in further detail in the first part of the “Appendix” section. To answer your questions, we made the same assumptions "in general and in the final model” and have considered “one-time” measurements. Linear regression is not suitable for our analysis since we consider a binary response (i.e., quit/not quit), as explained in the last paragraph of the “Data” section. Instead, we have adopted a generalized linear regression model, an extension of linear regression for binary classification. We have also developed more advanced predictive models with random forest, gradient boosting machines, and extreme gradient boosting algorithms. We conducted a longitudinal analysis by developing two-wave transitions in our paper; we used waves 1 and 2 for developing our model (not multiple waves), and after predictive model development (using waves 1- 2 cohort), we validated our results by applying waves 2-3 cohort. More specifically, we considered current smokers in wave 1 (baseline) and tracked smoking cessation for those respondents in wave 2 (follow-up) in order to develop a model capable of predicting smoking cessation for current smokers within a year, similar to what would likely be done in a smoking cessation clinical trial. As explained in the “Machine learning predictive models” section, we have ensured “data quality” mainly by data cleaning (lines 80-114 and 437-489) and measured model performance using a testing dataset. In response to this comment, we revised the following parts of the manuscript:

We added the following sentence to the “Data” section (pages 2-3, lines 54-57): 

We conducted a longitudinal analysis of PATH data in our study with one-time measurements. The same assumptions (as described below) are considered throughout the data cleaning and model development steps.

Page 3, lines 80-82 of the manuscript were revised as follows: 

Data cleaning is the process of removing (or fixing) incorrect, irrelevant, corrupted, incorrectly formatted, incomplete, or duplicate data to ensure data quality.

The following sentence was added to page 5 lines 170-172 of the manuscript:

We developed classification models with Generalized Linear Regression (GLM) (an extension of the linear regression for binary classification), RF, GBM, and extreme gradient boosting (XGBoost) algorithms.

Comment 8: How did the ethical considerations secure? Did the national or local IRB approval?

Response: We thank the reviewer for their question. As explained in the “Data” section, the need for IRB and participants’ consent were waived in our research since we used the open-access PATH dataset [9] (not the restricted version) in which all data were fully anonymized. We revised the “Data” section (page 3, lines 57-60) as follows:

We used the open-access PATH dataset (not the restricted version) [9], in which all data were fully deidentified. Therefore, Georgetown University and the University of Michigan Institutional Review Boards exempted our analysis from review.

Comment 9: The analysis needs further explanation.

Response: We thank the reviewer for their comment, and hope that the added explanations to the manuscript help make the analysis clearer. We edited the manuscript entirely, and revised the following parts:

“Data” section, pages 2-3, lines 54-57: 

We conducted a longitudinal analysis of PATH data in our study with one-time measurements. The same assumptions (as described below) are considered throughout the data cleaning and model development steps.

“Data cleaning” section, page 3, lines 80-82: 

Data cleaning is the process of removing (or fixing) incorrect, irrelevant, corrupted, incorrectly formatted, incomplete, or duplicate data to ensure data quality.

“Variable importance and direction effect” section, page 5, lines 157-161: 

The TreeSHAP is a typical method for variable interpretation, specifically in the public health domain. It has been used in tobacco research,[6] cancer prevention and control research,[7] and other applications.[8] The TreeSHAP analysis is used to explain the prediction of the machine learning models independently by each variable included in the model.

“Machine learning predictive models” section, page 5, lines 170-172:

We developed classification models with Generalized Linear Regression (GLM) (an extension of the linear regression for binary classification) RF, GBM, and extreme gradient boosting (XGBoost) algorithms.

“Machine learning predictive models” section, pages 5-6, lines 185-191:

The performance of the trained classifiers was compared based on classification accuracy (the ability to make correct predictions), sensitivity (the ability to predict “quit” cases correctly), specificity (the ability to predict “not quit” cases correctly), and the area under the receiver operating characteristic curve (AUC-ROC), (the ability to make correct predictions) in the testing set.

Comment 10: Ensure the completeness of the contents of the manuscript, E.g. The title lacks time period.

Response: We thank the reviewer for noting this. We have revised the title of the manuscript to: "Machine learning application for predicting smoking cessation among US adults: an analysis of Waves 1-3 of the PATH study”. Additionally, we have revised and edited the manuscript entirely to ensure the completeness of the content.

References

1. Population Assessment of Tobacco and Health (PATH) Study [United States] Public-Use Files (ICPSR 36498). Available from: https://www.icpsr.umich.edu/web/NAHDAP/studies/36498.

2. WHO sustainable development goals. Available from: https://www.who.int/europe/about-us/our-work/sustainable-development-goals/targets-of-sustainable-development-goal-3.

3. Lin, H., et al., National survey of smoking cessation provision in China. Tobacco induced diseases, 2019. 17.

4. Shaik, S.S., et al., Tobacco use cessation and prevention–A Review. Journal of clinical and diagnostic research: JCDR, 2016. 10(5): p. ZE13.

5. Jha, P., et al., 21st-century hazards of smoking and benefits of cessation in the United States. New England Journal of Medicine, 2013. 368(4): p. 341-350.

6. Medina, I.C. and M. Mohaghegh. Explainable Machine Learning Models for Prediction of Smoking Cessation Outcome in New Zealand. in 2022 14th International Conference on COMmunication Systems & NETworkS (COMSNETS). 2022. IEEE.

7. Inoguchi, T., et al., Association of serum bilirubin levels with risk of cancer development and total death. Scientific reports, 2021. 11(1): p. 1-12.

8. Shakeri, E., et al. Using SHAP Analysis to Detect Areas Contributing to Diabetic Retinopathy Detection. in 2022 IEEE 23rd International Conference on Information Reuse and Integration for Data Science (IRI). 2022. IEEE.

9. Population Assessment of Tobacco and Health (PATH) Study.; Available from: https://www.icpsr.umich.edu/web/NAHDAP/studies/36231.

---

## [Editor Report · Decision Letter 1]

25 May 2023

Machine learning application for predicting smoking cessation among US adults: an analysis of Waves 1-3 of the PATH study

PONE-D-23-00290R1

Dear Dr. Issabakhsh,

We’re pleased to inform you that your manuscript has been judged scientifically suitable for publication and will be formally accepted for publication once it meets all outstanding technical requirements.

Kind regards,

Mohammad Amin Fraiwan

Academic Editor

PLOS ONE
---

## [Editor Report · Acceptance letter]

31 May 2023

PONE-D-23-00290R1 

Machine learning application for predicting smoking cessation among US adults: an analysis of Waves 1-3 of the PATH study 

Dear Dr. Issabakhsh:

I'm pleased to inform you that your manuscript has been deemed suitable for publication in PLOS ONE. Congratulations! Your manuscript is now with our production department. 

Kind regards, 

on behalf of

Dr. Mohammad Amin Fraiwan 

Academic Editor

PLOS ONE